# Predictors for liver fibrosis in non-alcoholic patients with psoriatic diseases: A multicenter cross sectional-study

Norberto Carlos Chavez-Tapia[1]*, Beatriz A. Sanchez-Jimenez[2], Desiree Vidaña-Perez[3], Beatriz Corrales-Rosas[4], Brenda Balderas-Garces[1], Diana Vera-Izaguirre[5], Fermin Jurado Santa Cruz[6], Cesar Maldonado-Garcia[6], Eva Juarez-Hernandez[1], Misael Uribe[1]

1 Obesity and Digestive Diseases Unit and Translational Research Unit, Medica Sur Clinic & Foundation, Mexico City, Mexico, 2 Endoscopy Department, National Institute of Cancer, Mexico City, Mexico, 3 Center for Population Health Research, National Institute of Public Health, Cuernavaca, Mexico, 4 Hospital de Especialidades, Centro Médico Nacional Siglo XXI Instituto Mexicano del Seguro Social, Mexico City, Mexico, 5 Dermatology Department, Dr. Manuel Gea González General Hospital, Mexico City, Mexico, 6 Dr. Ladislao de la Pascua" Dermatologic Center, Secretaria de Salud del Distrito Federal, Mexico City, Mexico

* nchavezt@medicasur.org.mx

**Data Availability Statement:** All relevant data are within the manuscript and its Supporting Information files.

## Abstract

Psoriasis has been related to metabolic dysfunction-associated fatty liver disease and, liver fibrosis. This study aimed to evaluate the prevalence of liver fibrosis in psoriasis and identify predictors for fibrosis. This is a cross-sectional study conducted from December 2012 to June 2016 assessing psoriasis and psoriatic arthritis patients attended at four centers in Mexico City. Data regarding history of the skin disease, previous and current medication, and previously diagnosed liver disease was collected. Liver fibrosis was assessed with four different non-invasive methods (FIB4, APRI, NAFLD score and elastography). We compared data based on the presence of fibrosis. Adjusted-logistic regression models were performed to estimate OR and 95% CI. A total of 160 patients were included. The prevalence of significant fibrosis using elastography was 25% (n = 40), and 7.5% (n = 12) for advanced fibrosis. Patients with fibrosis had higher prevalence of obesity (60% vs 30.8%, P = 0.04), type 2 diabetes (40% vs 27.5%, P = 0.003), gamma-glutamyl transpeptidase levels (70.8 ±84.4 vs. 40.1±39.2, P = 0.002), and lower platelets (210.7±58.9 vs. 242.8±49.7, P = 0.0009). Multivariate analysis showed that body mass index (OR1.11, 95%CI 1.02–1.21), type 2 diabetes (OR 3.44, 95%CI 1.2–9.88), and gamma-glutamyl transpeptidase (OR 1.01, 95%CI1-1.02) were associated with the presence of fibrosis. The use of methotrexate was not associated. Patients with psoriasis are at higher risk of fibrosis. Metabolic dysfunction, rather than solely the use of hepatotoxic drugs, likely plays a major role; it may be beneficial to consider elastography regardless of the treatment used. Metabolic factors should be assessed, and lifestyle modification should be encouraged.

**Funding:** The authors received no specific funding for this work.

**Competing interests:** The authors have declared that no competing interests exist.

## Introduction

Psoriasis is a common chronic immune-mediated inflammatory skin disease that extends beyond aesthetic implications. This skin disease negatively affects the quality of life, particularly in Hispanics. Psoriasis has been associated with other components of the metabolic syndrome such as type 2 diabetes mellitus, high blood pressure, obesity, high cholesterol levels, cardiovascular disease, metabolic associated fatty liver disease (MAFLD), and psoriatic arthritis (PsA) [1–3]. The treatment of psoriasis is heterogeneous and includes topical administered drugs (retinoids, corticosteroids), phototherapy, serotonergic drugs, and some chemotherapeutic agents (i.e. methotrexate, retinoids) [4].

Liver damage in psoriasis was first thought to be caused by the use of hepatotoxic drugs such as methotrexate. However, current evidence shows that systemic comorbidities may have a greater impact on liver damage than drugs such as methotrexate. Psoriasis has been related to metabolic dysfunction-associated fatty liver disease and, liver fibrosis [3]; thus, hepatic monitoring is mandatory and should be based on risk factors for liver disease and the drug used for treatment [5].

Traditionally ultrasound, transaminases, and liver biopsy were the cornerstone to identify liver toxicity in patients with psoriasis under treatment with hepatotoxic drugs. Recently non-invasive methods for assessing liver fibrosis have been included in clinical guidelines for monitoring drug liver toxicity [5]. The use of transient elastography suggests a high proportion of patients with psoriasis and liver fibrosis; 14% of patients with significant fibrosis, and 8% with advanced fibrosis [6]. However, most of the information derived from studies with reduced sample sizes and did not include high-risk populations MAFLD, such as Hispanics. This study aimed to evaluate the prevalence of liver fibrosis in psoriasis and identify predictors for liver fibrosis in non-alcoholic patients with psoriatic diseases.

## Methods

### Study design and population

A cross-sectional study conducted from December 2012 to June 2016 assessing psoriasis and PsA patients attended at four dermatological centers in Mexico City (Dermatology Department Centro Medico Siglo XXI, Dermatological Center Dr. Ladislao de la Pascua, Asociación Mexicana contra la Psoriasis and Dermatology Department Hospital General Dr. Manuel Gea Gonzalez). Patients were consecutively enrolled.

Inclusion criteria comprised: patients older than 18 years, irrespectively length of diagnostic with psoriatic disease, treatment received, and activity of the disease. Those patients with a diagnosis of acute or chronic hepatitis, liver cancer, other chronic liver conditions, excess of alcohol consumption (>20 g/day for women and >30 g/day for men), or inability to perform liver elastography were excluded.

### Sample size

We used the formula for a dichotomous endpoint in a one-sample study. We considered an incidence of liver fibrosis of 3% in the general population and of 8% in patients with psoriasis, based on previously published studies [7, 8]. We considered an α of 0.05, β of 0.20 (statistical power of 80%). The required sample size was 127 participants.

### Clinical assessment

A specialized dermatologist confirmed the diagnosis of psoriasis or PsA. Clinical data such as smoking, history of liver disease, use of antiretrovirals and previous use of hepatotoxic drugs,

type 2 diabetes, high blood pressure, high cholesterol levels, were recruited. Biochemical data included serum aspartate aminotransferase (AST), alanine aminotransferase (ALT), total bilirubin, albumin, gamma-glutamyl transpeptidase (GGT), platelets, total cholesterol, and triglycerides. All clinical and biochemical data were recorded within a month previous to elastography measurement.

## Fatty liver

The diagnosis of fatty liver was performed using the Hepatic Steatosis Index, according to the formulae 8 x ALT/AST + BMI (+ 2 if type 2 diabetes yes, + 2 if female), the cut-off value was > 36 [9].

## Noninvasive liver fibrosis

Noninvasive liver fibrosis was assessed based on four different methods:

1. APRI: (AST (U/L)/ ULN) × 100/ platelet (109/L), with a cutoff value for advanced fibrosis of >1 [10].

2. FIB-4: (age (years) × AST (U/L))/ (platelets (109/L) × ALT (109/L)1/2), with a cutoff value for advanced fibrosis of >3.25 [11].

3. NAFLD Fibrosis Score: 1.675 + 0.037 × age (years) + 0.094 × Body Mass Index (BMI) (kg/m$^2$) + 1.13 × abnormal fasting glucose level or diabetes (yes = 1, no = 0) + 0.99 × AAR– 0.013 × number of platelets (×10$^9$/l)– 0.66 × albumin concentration (g/dl) with a cutoff value for advanced fibrosis of >0.676 [12].

4. Liver transient elastography (Fibroscan® Echosens, Paris, France) was realized according to manufacturer recommendations, from single measurement performed by a single experienced operator (>500 explorations). The cut-off to select a probe was ≥35 mm from skin to liver measured with abdominal sonography for the XL probe and <35 mm for the M probe. Fibrosis scores were assessed with the criteria proposed by Wong, et al: F2 >7.0 kPa, F3 >8.7 kPa, and F4 >10.3 kPa; clinically significant fibrosis was defined as fibrosis stage ≥ 2 and advanced fibrosis was defined as F3 or F4 [13]. At the time we started the study we did not have access to CAP measurements.

## Psoriatic diseases

Psoriasis phenotype, time since diagnosis, and previous and current treatments were evaluated. In patients with current or previous use of methotrexate, data were obtained regarding months of exposure, mean week dose, and total accumulated dose.

## Data analysis

For descriptive analysis, we estimated frequency and percentage for categorical variables. We used the Kolmogorov-Smirnov test to test the normality of the data, since nonparametric distribution exists, we reported continuous data with median and interquartile range.

We test the difference between patients with fibrosis and without fibrosis using Chi-squares for proportions and performed Mann-Whitney U test to compare continuous outcomes.

Logistic regression models were performed to estimate the odds ratio and 95% CI of the factors associated with the development of hepatic fibrosis. The logistic model was adjusted by age, years living with the disease, BMI, comorbidities, and other relevant biomarkers. Analyses were carried out using Stata 14 (Stata Corporation, TX 77845, USA).

The study was approved by the Ethics Committee of Médica Sur Clinic & Foundation and was conducted according to Good Clinical Practice Guidelines, and ethical principles based on the World Medical Association-Helsinki Declaration. Written informed consent was collected from each patient.

## Results

A total of 160 patients were included, being predominantly male (n = 100, 62.5%), with a median age of 54 [IQR 45–63] years, and a median BMI of 28.4 [25.7–32.2] kg/m$^2$. Regarding the comorbidities, smoking, type 2 diabetes mellitus, high blood pressure, and high cholesterol levels were observed in 40 (25%), 37 (23.1%), 47 (29.4%), and 60 (37.5%) patients respectively (Table 1).

The median length of psoriasis diagnosis was 15 years [IQR 10–22], the majority of patients had been treated with methotrexate (n = 137, 85.6%), and 97 (60.6%) subjects were currently treated with methotrexate, the mean week dose and accumulated dose were 12.4 mg [10–15] and 1250 mg [635–2845.7], respectively. Other frequent treatments were topic treatment (n = 21, 13.1%), adalimumab (n = 21, 13.1%), infliximab (n = 17, 10.6%), methotrexate plus keratolytics (n = 17, 10.6%), etarnecept (n = 11, 6.9%) and keratolytics (n = 10, 9.3%) (Table 1).

The laboratory work-up showed AST (UI/L) 27 [21–31], ALT (UI/L) 28 [20–37], total bilirubin (mg/dL) 0.65 [0.48–0.83], albumin (mg/dL) 4.4 [4.1–4.6], GGT (UI/L) 30 [23–47], and platelets 233 X 10$^9$/L [202–270]. Hepatic steatosis was diagnosed in 113 (70.6%) patients. Liver fibrosis was detected in 42 (26.3%) considering any noninvasive method used, and the liver stiffness (kPa) was 5.3 [4.2–7.1].

The prevalence of liver fibrosis using transient elastography was 25% (n = 40) for significant fibrosis, and 7.5% (n = 12) for advanced fibrosis. Advanced liver fibrosis considering other methods was 3.1% (n = 5), 4.4% (n = 7) and 3.1% (n = 5); for NAFLD score, APRI and FIB-4 respectively (Fig 1).

Those patients with significant fibrosis detected using transient elastography have higher prevalence of obesity (60% vs 30.8%, P = 0.04), type 2 diabetes mellitus (40% vs 27.5%, P = 0.014), higher BMI (31.01 [26.71–35.01] vs 27.76 [25.63–31.57] P = 0.006), GGT levels (41 [29–79.5] vs 29 [21.25–42.75] P = 0.002), lower platelets (210 [175–246.25] vs 239 [209.25–277.5] P = 0.001), NAFLD fibrosis score was higher (1.22 [0.92–1.48] vs 0.27 [-1.12–0.42], P = 0.002), and steatosis measured as hepatic steatosis index was also higher (41.6 [38.1–46.6] vs 37.7 [34.4–42.5], P = 0.002) (Table 2). The multivariate analysis showed that BMI (OR1.11, 95%CI 1.02–1.21), type 2 diabetes (OR 3.44, 95%CI 1.2–9.88), and GGT levels (OR 1.01, 95% CI1-1.02), were independently associated with the presence of significant fibrosis measured by elastography (Table 3).

## Discussion

Our aim was to evaluate the prevalence of liver fibrosis in psoriasis and to identify predictors for liver fibrosis in non-alcoholic patients with psoriatic diseases. Consistent with other similar studies, we found that 25% of patients had significant liver fibrosis using transient elastography, and 7.5% had advanced fibrosis [8, 14]. We observed a high prevalence of metabolic dysfunction-associated diseases such as diabetes mellitus type 2, high blood pressure, and high cholesterol levels. Hepatic steatosis was diagnosed in 70% of our population sample. Those patients with fibrosis detected using transient elastography have higher prevalence of obesity, type 2 diabetes mellitus, and higher GGT, NAFLD fibrosis score and hepatic steatosis index levels.

**Table 1. Clinical and demographic characteristics of patients.**

| Characteristic | % (n)/M [IQR] |
|---|---|
| *General characteristics* | |
| **Male sex** | 62.5% (100) |
| **Age (years)** | 54 [45–63] |
| BMI (kg/m$^2$) | 28.4 [25.7–32.2] |
| **Smoking** | 25% (40) |
| **Diabetes Mellitus** | 23.1% (37) |
| **Hypertension** | 29.4% (47) |
| **Dyslipidemia** | 37.5% (60) |
| **Total Cholesterol (mg/dl)** | 194 [170–213] |
| **Triglycerides (mg/dl)** | 170 [119–213] |
| *Psoriasis characteristics* | |
| **Time evolution of psoriasis (years)** | 15 [10–22] |
| **Treatment** | |
| **Methotrexate** | 23.8% (38) |
| **Topical treatment** | 13.1% (21) |
| **Adalimumab** | 13.1% (21) |
| **Infliximab** | 10.6% (17) |
| **Methotrexate + keratolytics** | 10.6% (17) |
| **Etanercept** | 6.9% (11) |
| **Keratolytics** | 6.3% (10) |
| **Vitamin D3 analogs** | 4.4% (7) |
| **Cyclosporine A** | 3.1% (5) |
| **Adalimumab + Methotrexate** | 2.5% (4) |
| **Acitretin** | 0.6% (1) |
| **Methotrexate + Other** | 0.6% (1) |
| **Chemoprophylaxis (isoniazid)** | 17.5% (28) |
| **Previous us of Methotrexate** | 85.6% (137) |
| **Current use of Methotrexate** | 60.6% (97) |
| **Months of exposure** | 37 [18–84] |
| **Median weekly dose (mg)** | 12.4 [10–15] |
| **Total accumulated dose (mg)** | 1250 [635–2845.7] |
| *Liver-related characteristics* | |
| **History of liver disease** | 10% (16) |
| **Previous hepatotoxic drug** | 2.5% (4) |
| **AST (UI/L)** | 27 [21–31] |
| **ALT (UI/L)** | 28 [20–37] |
| **TB (mg/dl)** | 0.65 [0.48–0.83] |
| **Albumin (mg/dl)** | 4.4 [4.1–4.6] |
| **GGT (UI/L)** | 30 [23–47] |
| Platelets (x10$^9$/L) | 233.5 [202–270] |
| **Steatosis (HIS)** | 70.6% (113) |
| **Liver fibrosis (any method)** | 26.3% (42) |
| **Fib4** | 1.05 [0.78–1.5] |
| **APRI** | 0.28 [0.20–0.38] |
| **NAFLD Fibrosis Score** | |
| **Without fibrosis** | 61.3% (98) |
| **Indeterminate** | 35.6% (57) |

*(Continued)*

**Table 1.** (Continued)

| Characteristic | % (n)/M [IQR] |
|---|---|
| Advanced fibrosis | 3.1% (5) |
| Liver Stiffness (kPa) | 5.3 [4.2–7.1] |

BMI Body Mass Index; AST Aspartate aminotransferase; ALT Alanine aminotransferase; TB total bilirubin; GGT gamma glutamyl transpeptidase; HIS Hepatic steatosis index; APRI AST to Platelet Ratio Index; NAFLD non-alcoholic fatty liver disease

Our data suggest an association of BMI, type 2 diabetes, and GGT levels with the presence of fibrosis measured by elastography. Furthermore, we found that for every unit of increase in BMI, the probability of having any type of fibrosis increases 11%, which is critical in countries like Mexico where the prevalence of overweight and obesity is more than 70%. [15]. It is well established that overweight and diabetes increase the risk and severity of liver fibrosis [16], and recently MAFLD has become the most frequent cirrhosis etiology in our country found in up to 30% of patients. [17] In Mexico diabetes prevalence increased form 9.2% in 2012 to 10.3% in 2018 [15]. However, in our population sample 23% had diabetes, and its association with other components of metabolic disfunction such as liver fibrosis places them at special risk of cardiovascular diseases [18], however its association with psoriatic diseases is poorly describe.

It is not entirely defined why some patients with psoriasis develop liver disease. First studies proposed a prime role of hepatotoxic drugs used to treat the disease. However, in our study population the use of methotrexate was not associated with the presence of fibrosis, supporting other published studies [19]. It has become increasingly apparent that there is a metabolic component in the development of liver fibrosis and, ultimately, cirrhosis [14, 20]. Sedentary lifestyle, high-fat and fructose diet, genetic and epigenetic factors may interact, leading to an excess of proinflammatory cytokines, mitochondrial dysfunction, and oxidative stress. Poly-morphisms in genes such as PNPLA3, TM6SF2, and GCKR promote steatosis through

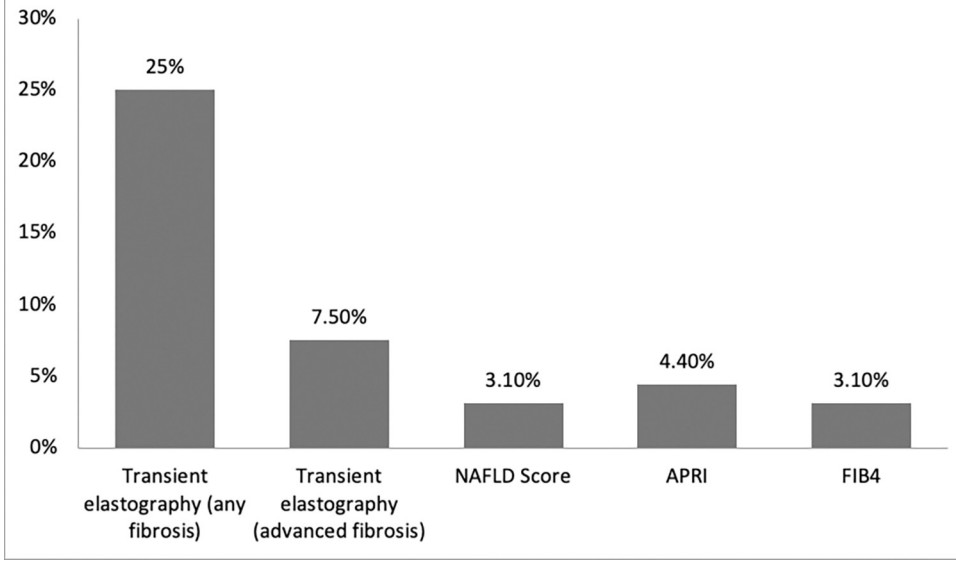

**Fig 1. Prevalence of liver fibrosis by noninvasive methods in patients with psoriasis.**

**Table 2. Comparison between patients with and without significant fibrosis.**

| | Patients without fibrosis (n = 120) | Patients with fibrosis (n = 40) | p-value |
|---|---|---|---|
| Sex (male %) | 65.8 | 52.5 | 0.131 |
| Age (mean ± SD) | 54.8 ± 13.1 | 52.9 ± 12.9 | 0.418 |
| BMI (%) | | | |
| Normal | 19.2 | 7.5 | 0.004 |
| Overweight | 50 | 32.5 | |
| Obesity | 30.8 | 60 | |
| Smoking (%) | 27.5 | 17.5 | 0.206 |
| Type 2 diabetes (%) | 17.5 | 40 | 0.003 |
| Hypertension (%) | 27.5 | 35 | 0.367 |
| Dyslipidemia (%) | 38.3 | 35 | 0.706 |
| Chemoprophylaxis (%) | 20.5 | 12.8 | 0.286 |
| Methotrexate (%) | 67.3 | 56.8 | 0.246 |
| Albumin mg/dl (mean ± SD) | 4.3 [3.9–4.6] | 4.4 [4.2–4.6] | 0.266 |
| GGT UI/L (mean ± SD) | 41 [29–79.5] | 29 [21.25–42.75] | 0.002 |
| Platelets x10$^9$/L (mean ± SD) | 210 [175–246.25] | 239 [209.25–277.5] | 0.001 |
| Cholesterol mg/dl (mean ± SD) | 192.5 [162–207] | 194[174–216.75] | 0.166 |
| Tryglicerides mg/dl (mean ± SD) | 180 [142.7–213] | 163.5 [115.2–2112.5] | 0.148 |
| Years with disease (mean ± SD) | 15 [8–21.7] | 15 [10.5–25] | 0.314 |
| NAFLD Score (mean ± SD) | 1.22 [0.92–1.48] | 0.27 [-1.12–0.42] | 0.002 |
| HSI (mean ± SD) | 41.6 [38.1–46.6] | 37.7 [34.4–42.5] | 0.002 |

*Chi-squares for proportion and t test for means and standard deviations. BMI Body Mass Index; GGT gamma glutamyl transpeptidase; NAFLD non-alcoholic fatty liver disease; HIS Hepatic steatosis index.

interaction with distinct metabolic mechanisms, predisposing to metabolic dysfunction-associated fatty liver disease [21]. Another potential explanation is the lower levels of serum adiponectin in patients with psoriasis [22], a known risk factor for liver steatosis in our population [23].

**Table 3. Multivariate analysis for significant of fibrosis (elastography).**

| | OR (CI 95%) | P value |
|---|---|---|
| Sex (male) | 0.7 (0.29, 1.73) | 0.445 |
| BMI | 1.11 (1.02, 1.21) | 0.019 |
| Years w/disease | | |
| < 5 | REF | |
| 5 to 15 | 3.13 (0.69, 14.13) | 0.139 |
| 15 + | 2.99 (0.61, 14.68) | 0.178 |
| Albumin (mg/dl) | 0.65 (0.26, 1.61) | 0.349 |
| Platelets (x10$^{9/1}$) | 1.94 (0.71, 5.25) | 0.194 |
| Diabetes | 3.44 (1.2, 9.88) | 0.022 |
| Hypertension | 1.2 (0.42, 3.39) | 0.735 |
| GGT (UI/L) | 1.01 (1, 1.02) | 0.011 |
| Methotrextate | 0.97 (0.23, 4.07) | 0.964 |

*Adjusted by age, cholesterol, triglycerides, chemoprophylaxis, and dyslipidemia. BMI body mass index; GGT gamma glutamyl transpeptidase.

There are some limitations to this study that we need to acknowledge. First, this is a cross-sectional study, and it was not designed to evaluate causality. However, we were able to provide information about the association of important variables with our outcome. Second, we did not assess relevant parameters associated with metabolic syndrome, such as HOMA-IR or the severity of the disease, but we controlled by other covariates like BMI, years living with the disease and treatments used. Third, we lacked a liver biopsy however the diagnosis of liver fibrosis was assessed by four different ways, and lastly, at the time we conducted the study we did not have access to CAP measurements.

Some strengths in this study are the large sample size; to our knowledge, only two studies with a larger sample size assessing liver fibrosis with transient elastography in patients with psoriasis had been published [8, 24]. We did not find any study that included high-risk population such as Hispanics.

In conclusion, patients with psoriasis are at higher risk of liver fibrosis. It has become increasingly clear that there is a metabolic component in the development of liver fibrosis rather than solely the use of hepatotoxic drugs such as methotrexate. Therefore, it may be beneficial to consider transient elastography regardless of the treatment used. Also, metabolic factors should be assessed cautiously in the management of a patient with psoriasis, and strategies to help to modify lifestyle should be strongly encouraged. Larger prospective controlled trials at low risk of bias should be conducted to support our findings.

## Supporting information

**S1 Database.**
(SAV)

## Author Contributions

**Conceptualization:** Norberto Carlos Chavez-Tapia.

**Data curation:** Eva Juarez-Hernandez.

**Formal analysis:** Norberto Carlos Chavez-Tapia, Beatriz A. Sanchez-Jimenez, Desiree Vidaña-Perez.

**Investigation:** Beatriz Corrales-Rosas, Brenda Balderas-Garces, Diana Vera-Izaguirre, Fermin Jurado Santa Cruz, Cesar Maldonado-Garcia, Misael Uribe.

**Methodology:** Norberto Carlos Chavez-Tapia.

**Project administration:** Norberto Carlos Chavez-Tapia.

**Resources:** Beatriz Corrales-Rosas, Diana Vera-Izaguirre, Fermin Jurado Santa Cruz, Cesar Maldonado-Garcia, Misael Uribe.

**Writing – original draft:** Beatriz A. Sanchez-Jimenez.

**Writing – review & editing:** Beatriz A. Sanchez-Jimenez, Desiree Vidaña-Perez.

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
