## [Decision Letter · Decision Letter 0]

8 Feb 2022

PONE-D-21-38265Predictors for liver fibrosis in non-alcoholic patients with psoriatic diseases: a multicenter cross sectional-studyPLOS ONE

Dear Dr. Chavez-Tapia,

Thank you for submitting your manuscript to PLOS ONE. After careful consideration, we feel that it has merit but does not fully meet PLOS ONE’s publication criteria as it currently stands. Therefore, we invite you to submit a revised version of the manuscript that addresses the points raised during the review process.

We look forward to receiving your revised manuscript.

Kind regards,

Daisuke Tokuhara

Academic Editor

PLOS ONE

Journal Requirements:

2. Please provide additional details regarding participant consent. In the Methods section, please ensure that you have specified (1) whether consent was informed and (2) what type you obtained (for instance, written or verbal). If your study included minors, state whether you obtained consent from parents or guardians. If the need for consent was waived by the ethics committee, please include this information.

Additional Editor Comments:

Thank you for submitting your interesting research paper to PLOS One. Reviewers provided constructive comments. Based on reviewers' comments, we ask you to respond to their comments sincerely and submit the revised manuscript.

Reviewers' comments:

Reviewer's Responses to Questions

**Comments to the Author**

1. Is the manuscript technically sound, and do the data support the conclusions?

Reviewer #1: No

Reviewer #2: Partly

Reviewer #3: Yes

Reviewer #4: No

2. Has the statistical analysis been performed appropriately and rigorously? 

Reviewer #1: Yes

Reviewer #2: No

Reviewer #3: Yes

Reviewer #4: Yes

3. Have the authors made all data underlying the findings in their manuscript fully available?

Reviewer #1: No

Reviewer #2: Yes

Reviewer #3: Yes

Reviewer #4: Yes

4. Is the manuscript presented in an intelligible fashion and written in standard English?

Reviewer #1: Yes

Reviewer #2: No

Reviewer #3: Yes

Reviewer #4: Yes

5. Review Comments to the Author

Reviewer #1: Psoriasis has been related to metabolic dysfunction-associated fatty liver disease and liver fibrosis. This study aimed to evaluate the prevalence of liver fibrosis in psoriasis and identify predictors for fibrosis.

Total 160 patients were included in the study. The prevalence of significant fibrosis and advanced fibrosis using elastography was 25% (n=40), and 7.5% (n=12) respectively. Patients with advanced fibrosis had higher prevalence of obesity (60% vs 30.8%, P=0.04), type 2 diabetes (40% vs 27.5%, P=0.003), gamma-glutamyl transpeptidase levels (70.8±84.4 vs. 40.1±39.2, P=0.002), and lower platelets (210.7±58.9 vs. 242.8±49.7, P=0.0009) compared to the non-fibrosis patients.

Multivariate analysis showed that body mass index, type 2 diabetes, and gamma-glutamyl transpeptidase were associated with the presence of fibrosis rather than methotrexate.

Sample size was enough. So, the study was adequately powered.

The non-invasive methods those were used to identify steatosis and advanced fibrosis are now externally validated and used worldwide.

All the non-invasive methods including transient elastography used the standard cutoff value to include or exclude advanced fibrosis.

On contrary to the common belief (liver fibrosis due to methotrexate toxicity in psoriasis), this study opened a new dimension that drug association isn’t statistically significant, rather components of metabolic syndrome are the main risk factors for advanced liver fibrosis.

All the data were analyzed with appropriate testing methods such as chi square test, student t test, logistic regression method, univariate and multivariate analysis.

This study also revealed the importance of TE to detect advanced liver fibrosis in psoriasis irrespective of disease activity, duration and dose of methotrexate.

MAOJR REVIEW

Table 1 is not relevant and not understandable

Table 2 and description of results are not similar. Table described presence/ absence of fibrosis but results described advance fibrosis

Only non-invasive methods were used to identify advanced fibrosis which could be further validated by the ‘gold standard’ liver biopsy.

This study didn’t further clarify the subsets of fibrosis population in terms of difference in the disease activity, length of duration of disease and differences in the dose and cumulative exposure to methotrexate in table 2 and 3 . If these data would be available, it would be lot easier to understand the main predictors for liver fibrosis.

Reviewer #2: 1. Abstract

a. Accordingly with submission guidelines distinct headings should not be applied within the abstract.

Results section

b. Line 34: Authors refer to transient elastography but did no other comment regarding other non-invasive methods. Maybe a sentence regarding other methods should be made.

2. Methods

Data analysis

a. Page 6, Lines 123 and 124

Authors stated that for continuous variables results are shown as median and interquartile range. A normality test should be performed prior to deciding the use of mean/standard deviation or median/IQR.

3. Results

Table 1. Baseline characteristics of the include patients.

a. This reviewer did not understand what are columns A from E presented in Table 1. Which groups are these?

b. Sample size described within groups(A-E) exceeds the 160 patients aforementioned (line 133).

c. Continuous variables are presented as mean and standard deviation. Authors previously stated continuous variables are presented as median and IQ. This should be reviewed.

d. Table 1 also states percentage of patients with Metabolic Syndrome. How did authors evaluate metabolic syndrome in enrolled patients?

Text

e. Lines 144 to 146: Authors describe frequency of topic treatment for psoriasis and refer to Table 1. This information is not showed in Table 1.

f. Authors described in methods that liver fibrosis would be assessed by 4 different non-invasive methods: APRI, FIB-4, NAFLD Fibrosis Score and Transient Elastography. In Results section (line 150) it is stated: “Liver fibrosis was detected in 42 (26.3%) considering any noninvasive method used”. By liver fibrosis they mean significant or advanced fibrosis? For APRI, FIB-4 and NAFLD Fibrosis Score are only presented cut off values for advanced fibrosis and not for significant fibrosis. Authors should be clear during the text and stick to the terms significant (≥F2) and advanced fibrosis ( ≥F3) to avoid misinterpretation.

g. Transient elastography presented higher rates of advanced liver fibrosis than other methods (lines 152 to 155). Tables 2 and 3 present results of comparison of fibrosis in patients using transient elastography as a determinant of groups, but the same analysis was not made for other methods. Did authors consider transient elastography the gold-standard? Maybe authors could state TE was considered the gold standard. It could be emphasized the differences encountered when patients are compared by means of significant and advanced liver fibrosis.

h. Did authors considered performing a linear regression using continuous values obtained from transient elastography (kPa) with other the analyzed variables?

i. Did authors have access to controlled attenuation parameter (CAP) measurements in transient elastography? Since this measurement could be useful to determine steatosis (Ref.: Castera L, Friedrich-Rust M, Loomba R.. Noninvasive assessment of liver disease in patients with nonalcoholic fatty liver disease. Gastroenterology. 2019;156:1264–1281.e4.)

Reviewer #3: This is a very interesting original paper regarding an increasingly frequent health issue in Latin populations.

The sample size is adequate although there is no "gold standard" available.

The conclusions are important as they reinforce the information that the severity of liver disease in psoriatic patients relies on their metabolic profile and not in their treatment history.

The paper is well written, fully comprehensible, with minor mistakes.

Some points I would like to highlight:

1) The authors evaluated the prevalence of liver fibrosis using elastography and noninvasive serum scores. The prevalence of advanced liver fibrosis was different according to the different methods (figure 1). The authors did not discuss these discording results. Also, it was provided with no explanation for choosing liver stiffness as reference.

2) There is scarcity of information regarding the evaluation of liver fibrosis: how experienced was the Fibroscan operator? Which probe was used, M or XL? Which was the criteria for choosing the probe?

3) It is well stablished that the EHT ma y vary intra and inter observatory. It would be interesting to have more than one liver fibrosis evaluation, as it was taken as the reference. If not available, it should be discussed.

4) Was not CAP (controlled attenuation parameter) available? It should also be discussed.

5) Steatosis was evaluated with HIS, which is not also a “gold standard” method. BMI, but not steatosis, was associated with liver fibrosis in the multivariate analysis. It should be discussed as they are correlated variables.

6) I could not understand table 1. What do the five groups named from A to E represent and what are those numbers ( 255, 266, 238)?

Reviewer #4: The manuscript presented is confusing and does not meet the objectives of the title. It is not clear wether patients with psoriasis worsened or not. If so, the statistical data do not show the worsening conclusion. The article focuses more on NASH patients. In addition , I did not understand table 1 regarding the groups presented. The authors showed a study of 160 patients, but the table shows 5 subgroups that are not mentioned in the work. The discussion is unrepresentative in this study. The authors discuss very little about the changes resulting from the treatment of psoriasis that might lead to liver damage

6. PLOS authors have the option to publish the peer review history of their article (what does this mean?). If published, this will include your full peer review and any attached files.

Reviewer #1: **Yes: **Shahinul Alam

Reviewer #2: No

Reviewer #3: No

Reviewer #4: No

---

## [Author Response · Author response to Decision Letter 0]

29 Aug 2022

Thank you for accepting to review our manuscript. We found every one of your comments valuable and we have made the corresponding changes that you kindly suggested. Comments and changes are detailed below: 

Reviewer #1: Psoriasis has been related to metabolic dysfunction-associated fatty liver disease and liver fibrosis. This study aimed to evaluate the prevalence of liver fibrosis in psoriasis and identify predictors for fibrosis.

Total 160 patients were included in the study. The prevalence of significant fibrosis and advanced fibrosis using elastography was 25% (n=40), and 7.5% (n=12) respectively. Patients with advanced fibrosis had higher prevalence of obesity (60% vs 30.8%, P=0.04), type 2 diabetes (40% vs 27.5%, P=0.003), gamma-glutamyl transpeptidase levels (70.8±84.4 vs. 40.1±39.2, P=0.002), and lower platelets (210.7±58.9 vs. 242.8±49.7, P=0.0009) compared to the non-fibrosis patients.

Multivariate analysis showed that body mass index, type 2 diabetes, and gamma-glutamyl transpeptidase were associated with the presence of fibrosis rather than methotrexate.

Sample size was enough. So, the study was adequately powered.

The non-invasive methods those were used to identify steatosis and advanced fibrosis are now externally validated and used worldwide.

All the non-invasive methods including transient elastography used the standard cutoff value to include or exclude advanced fibrosis.

On contrary to the common belief (liver fibrosis due to methotrexate toxicity in psoriasis), this study opened a new dimension that drug association isn’t statistically significant, rather components of metabolic syndrome are the main risk factors for advanced liver fibrosis.

All the data were analyzed with appropriate testing methods such as chi square test, student t test, logistic regression method, univariate and multivariate analysis.

This study also revealed the importance of TE to detect advanced liver fibrosis in psoriasis irrespective of disease activity, duration and dose of methotrexate.

MAJOR REVIEW

Table 1 is not relevant and not understandable

R: Table one was not a part of this article. We now included the right one.

Table 2 and description of results are not similar. Table described presence/ absence of fibrosis

R: We changed the description to “significant fibrosis” (referring to patients with TE ≥ 2).

Only non-invasive methods were used to identify advanced fibrosis which could be further validated by the ‘gold standard’ liver biopsy. 

R: We realize the lack of liver biopsy is a limitation, nevertheless, the use of non-invasive tests, such as the ones we used in our study is well documented “The advent of non-invasive tests (NITs) has now replaced the role of liver biopsy. These approaches can overcome the limitations of liver biopsy and have become more widely used in routine clinical practice. Imaging-based elastography and serum biomarkers are currently the two main NITs for liver fibrosis staging”) scientific reports;2022 12:4913 

This study didn’t further clarify the subsets of fibrosis population in terms of difference in the disease activity, length of duration of disease and differences in the dose and cumulative exposure to methotrexate in table 2 and 3 . If these data would be available, it would be lot easier to understand the main predictors for liver fibrosis.

R: We now include table 1 which classify patients according to length of duration of disease and use of methotrexate and other drugs. 

Reviewer #2: 1. Abstract

a. Accordingly with submission guidelines distinct headings should not be applied within the abstract.

R: We deleted the headings within the abstract. 

Results section

b. Line 34: Authors refer to transient elastography but did no other comment regarding other non-invasive methods. Maybe a sentence regarding other methods should be made.

R: We described on lines 167 to 169: “. Advanced liver fibrosis considering other methods was 3.1% (n=5), 4.4% (n=7) and 3.1% (n=5); for NAFLD score, APRI and FIB-4 respectively (Fig 1)”

2. Methods

Data analysis

a. Page 6, Lines 123 and 124

Authors stated that for continuous variables results are shown as median and interquartile range. A normality test should be performed prior to deciding the use of mean/standard deviation or median/IQR.

R: Kolmogorov test was performed to assess normality.

3. Results

Table 1. Baseline characteristics of the include patients.

a) This reviewer did not understand what are columns A from E presented in Table 1. Which groups are these?

b) Sample size described within groups(A-E) exceeds the 160 patients aforementioned (line 133).

c) Continuous variables are presented as mean and standard deviation. Authors previously stated continuous variables are presented as median and IQ. This should be reviewed.

d) Table 1 also states percentage of patients with Metabolic Syndrome. How did authors evaluate metabolic syndrome in enrolled patients? 

e) Lines 144 to 146: Authors describe frequency of topic treatment for psoriasis and refer to Table 1. This information is not showed in Table 1.

R: (a-e): table one was not a part of this article. We now included the right one. 

f. Authors described in methods that liver fibrosis would be assessed by 4 different non-invasive methods: APRI, FIB-4, NAFLD Fibrosis Score and Transient Elastography. In Results section (line 150) it is stated: “Liver fibrosis was detected in 42 (26.3%) considering any noninvasive method used”. By liver fibrosis they mean significant or advanced fibrosis? For APRI, FIB-4 and NAFLD Fibrosis Score are only presented cut off values for advanced fibrosis and not for significant fibrosis. Authors should be clear during the text and stick to the terms significant (≥F2) and advanced fibrosis ( ≥F3) to avoid misinterpretation.

R: We added in the methods section “…clinically significant fibrosis was defined as fibrosis stage ≥ 2 and advanced fibrosis was defined as F3 or F4”. TE has been widely used to assess any degree of fibrosis (F1 to F4), but biochemical scores are mainly use to assess advanced fibrosis. 

g. Transient elastography presented higher rates of advanced liver fibrosis than other methods (lines 152 to 155). Tables 2 and 3 present results of comparison of fibrosis in patients using transient elastography as a determinant of groups, but the same analysis was not made for other methods. Did authors consider transient elastography the gold-standard? Maybe authors could state TE was considered the gold standard. It could be emphasized the differences encountered when patients are compared by means of significant and advanced liver fibrosis.

R: TE has been extensively compared to other NITs and is considered the best predictor of significant fibrosis. 

h. Did authors considered performing a linear regression using continuous values obtained from transient elastography (kPa) with other the analyzed variables?

R: We performed linear regressions for all the variables we measured. We reported the most significant. 

i. Did authors have access to controlled attenuation parameter (CAP) measurements in transient elastography? Since this measurement could be useful to determine steatosis (Ref.: Castera L, Friedrich-Rust M, Loomba R.. Noninvasive assessment of liver disease in patients with nonalcoholic fatty liver disease. Gastroenterology. 2019;156:1264–1281.e4.)

R: At the time we started the study (2012) we did not have access to CAP measurements. We now include this limitation both on methods and on limitations. 

Reviewer #3: This is a very interesting original paper regarding an increasingly frequent health issue in Latin populations.

The sample size is adequate although there is no "gold standard" available.

The conclusions are important as they reinforce the information that the severity of liver disease in psoriatic patients relies on their metabolic profile and not in their treatment history.

The paper is well written, fully comprehensible, with minor mistakes.

Some points I would like to highlight:

1) The authors evaluated the prevalence of liver fibrosis using elastography and noninvasive serum scores. The prevalence of advanced liver fibrosis was different according to the different methods (figure 1). The authors did not discuss these discording results. Also, it was provided with no explanation for choosing liver stiffness as reference.

R: TE has been extensively compared to other NITs and is considered the best predictor of significant fibrosis. 

2) There is scarcity of information regarding the evaluation of liver fibrosis: how experienced was the Fibroscan operator? Which probe was used, M or XL? Which was the criteria for choosing the probe?

R: We added the following information on methods: “…performed by a single experienced operator (>500 explorations). The cut-off to select a probe was ≥35 mm from skin to liver measured with abdominal sonography for the XL probe and <35 mm for the M probe”.

3) It is well stablished that the EHT may vary intra and inter observatory. It would be interesting to have more than one liver fibrosis evaluation, as it was taken as the reference. If not available, it should be discussed.

R: We used IQR to standardize the quality of TE (IQR <0.3). Four different non-invasive tests (NITs) to assess liver fibrosis: TE, APRI, FIB4, and NAFLD fibrosis score. We realized that the lack of liver biopsy was a limitation, nevertheless the use of NITs is not only well documented but encourage in current practice since they seemed to overcome the limitations of liver biopsy (evaluate a greater volume of tissue and have less associated complications). (scientific reports;2022 12:4913)

4) Was not CAP (controlled attenuation parameter) available? It should also be discussed.

R: At the time we started the study (2012) we did not have access to CAP measurements. We now include this limitation both on methods and on limitations. . 

5) Steatosis was evaluated with HIS, which is not also a “gold standard” method. BMI, but not steatosis, was associated with liver fibrosis in the multivariate analysis. It should be discussed as they are correlated variables.

R: We are aware HIS is not a standard method to assess liver steatosis, furthermore it is not correlated with disease severity, therefore we thought it was understandable that it was somehow statistically different, but this association did not remain significant in the multivariate analysis. We are aware using CAP would have been the best tool, sadly we did not have access to it until the study was almost done so we could get this information only in a very small proportion of patients. 

6) I could not understand table 1. What do the five groups named from A to E represent and what are those numbers ( 255, 266, 238)?

R: Table one was not a part of this article. We now included the right one.

Reviewer #4: The manuscript presented is confusing and does not meet the objectives of the title. It is not clear wether patients with psoriasis worsened or not. If so, the statistical data do not show the worsening conclusion. The article focuses more on NASH patients. In addition , I did not understand table 1 regarding the groups presented. The authors showed a study of 160 patients, but the table shows 5 subgroups that are not mentioned in the work. The discussion is unrepresentative in this study. The authors discuss very little about the changes resulting from the treatment of psoriasis that might lead to liver damage

R: Table one was not a part of this article. We now included the right one.

---

## [Decision Letter · Decision Letter 1]

10 Nov 2022

PONE-D-21-38265R1Predictors for liver fibrosis in non-alcoholic patients with psoriatic diseases: a multicenter cross sectional-studyPLOS ONE

Dear Dr. Chavez-Tapia,

Thank you for submitting your manuscript to PLOS ONE. After careful consideration, we feel that it has merit but does not fully meet PLOS ONE’s publication criteria as it currently stands. Therefore, we invite you to submit a revised version of the manuscript that addresses the points raised during the review process.

Thanks for revising your work. Almost all of the peer-review comments are addressed in this version. There are some other concerns that should be addressed before being published.

- Did you test the quantitative variables regarding their normality of distribution? Please mention the name of the test in the statistical analysis section.

- Please also mention the tests that were used for evaluating the variables with and without a normal distribution in the statistical analysis section (e.g. t-test and its nonparametric equivalent).

- Please make sure to use the mean +/- standard deviation for the variables with a normal distribution and the median [IQR] for the variables without a normal distribution (in the result's text and tables).

- Please make sure to use parametric tests to evaluate the variables with normal distribution and non-parametric tests to evaluate the variables without a normal distribution (in the result's text and tables).

- The style/formatting of tables 2 and 3 need to be changed. Please change them to a simple plain style.

- The discussion part needs to be improved. Please use some recent studies to expand the discussion.

We look forward to receiving your revised manuscript.

Kind regards,

Hamidreza Karimi-Sari, MD

Academic Editor

PLOS ONE

Reviewers' comments:

Reviewer's Responses to Questions

**Comments to the Author**

1. If the authors have adequately addressed your comments raised in a previous round of review and you feel that this manuscript is now acceptable for publication, you may indicate that here to bypass the “Comments to the Author” section, enter your conflict of interest statement in the “Confidential to Editor” section, and submit your "Accept" recommendation.

Reviewer #1: All comments have been addressed

Reviewer #2: (No Response)

Reviewer #5: All comments have been addressed

2. Is the manuscript technically sound, and do the data support the conclusions?

Reviewer #1: Yes

Reviewer #2: Yes

Reviewer #5: Yes

3. Has the statistical analysis been performed appropriately and rigorously? 

Reviewer #1: Yes

Reviewer #2: Yes

Reviewer #5: Yes

4. Have the authors made all data underlying the findings in their manuscript fully available?

Reviewer #1: Yes

Reviewer #2: Yes

Reviewer #5: Yes

5. Is the manuscript presented in an intelligible fashion and written in standard English?

Reviewer #1: Yes

Reviewer #2: Yes

Reviewer #5: Yes

6. Review Comments to the Author

Reviewer #1: Author has responded sufficiently to review. Though discussion is too small. This manuscript may contribute in clinical decision making of psoriasis. MTX prescription would be justified with fibrosis in MAFLD patients with follow up of patients by TE

Reviewer #2: By TE, 7.5% of patients presented advanced fibrosis while in other methods the prevalence was lower. This difference was statically significant? Did all methods identify the same patients as presenting advanced fibrosis?

In table 1 variables are presented by median IQR and in table 2 they are presented by mean, standard deviation and a T test was performed. If variables have abnormal distribution, they should be presented by median and an ANOVA test should be performed. This seems to be the case for GGT.

Reviewer #5: (No Response)

7. PLOS authors have the option to publish the peer review history of their article (what does this mean?). If published, this will include your full peer review and any attached files.

Reviewer #1: **Yes: **Shahinul Alam

Reviewer #2: No

Reviewer #5: No

---

## [Author Response · Author response to Decision Letter 1]

11 Aug 2023

Dear reviewers,

We appreciate very much you agreed to evaluate our study, we already made the changes you suggested. The modifications are detailed below:

1. - Did you test the quantitative variables regarding their normality of distribution? Please mention the name of the test in the statistical analysis section.

R: We added in the data analysis section the following paragraph: “We used the Kolmogorov-Smirnov test to test the normality of the data, since nonparametric distribution exists, we reported continuous data with median and interquartile range”. 

2. Please also mention the tests that were used for evaluating the variables with and without a normal distribution in the statistical analysis section (e.g. t-test and its nonparametric equivalent).

 R: We added in the data analysis section the following: “We test the difference between patients with fibrosis and without fibrosis using Chi-squares for proportions and performed Mann-Whitney U test to compare continuous outcomes”. 

3. - Please make sure to use the mean +/- standard deviation for the variables with a normal distribution and the median [IQR] for the variables without a normal distribution (in the result's text and tables).

R: We changed the results that were reported with mean and SD with median an IQR in both the result section and the table. 

4. Please make sure to use parametric tests to evaluate the variables with normal distribution and non-parametric tests to evaluate the variables without a normal distribution (in the result's text and tables).

R: We changed the results that were reported with mean and SD with median an IQR in both the result section and the table. 

5. The style/formatting of tables 2 and 3 need to be changed. Please change them to a simple plain style.

R: We changed the style of tables 2 and 3. 

6. The discussion part needs to be improved. Please use some recent studies to expand the discussion.

R: We changed some sections on the discussion.

---

## [Editor Report · Decision Letter 2]

13 Aug 2023

Predictors for liver fibrosis in non-alcoholic patients with psoriatic diseases: a multicenter cross sectional-study

PONE-D-21-38265R2

Dear Dr. Chavez-Tapia,

We’re pleased to inform you that your manuscript has been judged scientifically suitable for publication and will be formally accepted for publication once it meets all outstanding technical requirements.

Kind regards,

Hamidreza Karimi-Sari

Academic Editor

PLOS ONE

---

## [Editor Report · Acceptance letter]

3 Apr 2024

PONE-D-21-38265R2 

PLOS ONE

Dear Dr. Chavez-Tapia, 

I'm pleased to inform you that your manuscript has been deemed suitable for publication in PLOS ONE. Congratulations! Your manuscript is now being handed over to our production team.

Kind regards, 

on behalf of

Hamidreza Karimi-Sari 

Academic Editor

PLOS ONE